# Molecular Mechanisms of Alfalfa Response to Abiotic Stresses

**DOI:** 10.3390/plants14030487

**Published:** 2025-02-06

**Authors:** Wenxin Peng, Wenqi Cai, Jieyi Pan, Xinru Su, Liru Dou

**Affiliations:** College of Grassland Science and Technology, China Agricultural University, Beijing 100193, China; pwx2784796543@163.com (W.P.); 19861590011@139.com (W.C.); 2022324010211@cau.edu.cn (J.P.); xinru_su2022@163.com (X.S.)

**Keywords:** alfalfa, abiotic stress, molecular mechanisms

## Abstract

Alfalfa (*Medicago sativa* L.), a high-quality perennial legume forage, is pivotal in global animal husbandry and ecological systems. However, its growth and production are threatened by various abiotic stresses, including drought, salinity, low temperatures, and heavy metal toxicity. This review summarizes recent research on the molecular mechanisms underlying alfalfa’s responses to these environmental adversities. It provides a theoretical foundation for enhancing the stress resistance of alfalfa, offering a valuable reference for breeding high-quality, stress-resistant alfalfa varieties.

## 1. Introduction

Alfalfa, known as the ‘Queen of Forage’, is a high-quality perennial legume forage with wide adaptability, strong stress resistance, high yield, good palatability, and high crude protein content. In China, alfalfa is widely planted in the arid and semi-arid regions in the North to meet the demand of the livestock feed market, which has high environmental and ecological value as well as economic and agricultural value [1]. However, under the dual impacts of global climate change and human activities, alfalfa is facing increasingly severe abiotic stresses such as drought [2], salinity [3], low temperature [4], and heavy metal pollution [5]. These abiotic stresses pose serious threats to the growth, yield, and quality of alfalfa [6], negatively impacting its applications in agricultural production.

In the context of the continuous evolution of the global ecological environment, abiotic stress is a key factor limiting the growth and survival of plants. When faced with abiotic stress, plants usually adopt a series of complex physiological, biochemical, and molecular mechanisms for response and adaptation [7]. Currently, the response mechanisms of alfalfa to abiotic stress have become a hot topic of research both domestically and internationally. An increasing number of studies are revealing the molecular mechanisms of alfalfa’s response to abiotic stress through a series of complex processes, from the perception and recognition of environmental signals, the activation of intracellular signal transduction pathways, to changes in gene expression regulation and the corresponding physiological and biochemical characteristics. Some key genes have been identified as playing crucial roles in this process. In this review, we systematically summarize the latest advancements in the response of alfalfa to abiotic stresses, including drought, salinity, low temperature, and heavy metal stress. This is of great significance for promoting the breeding of stress-tolerant alfalfa and enhancing its productivity under various environmental conditions.

## 2. Drought Stress

Drought stress is considered to be the most important factor limiting plant growth [8]. Under drought stress, alfalfa shows morphological characteristics such as decreased germination rate [9], slow seedling growth, increased root-crown ratio [10,11], and significantly reduced aboveground biomass [12]. Further studies revealed that alfalfa responds to drought stress by regulating photosynthesis, water metabolism, antioxidant system, and endogenous plant hormones [13], multi-omics analysis further revealed that most flavonoid compounds were significantly accumulated after drought stress to reduce oxidative damage, and there may be post-transcriptional regulation mechanisms and protein-protein interaction mechanisms related to flavonoid synthesis under drought stress that affect the chemical synthesis of flavonoids [14].

Signal sensing mechanisms and transduction pathways are key links in alfalfa’s ability to sense abiotic stress and initiate appropriate responses to maintain growth and survival. Alfalfa growth, development, and abiotic stress response are critically regulated by calcium signaling. The expression of the hypertonic calcium ion (Ca^2+^) receptor *MsOSCA* genes was found to be induced by drought stress [15]. Protein kinase-mediated signaling pathways are one of the important mechanisms by which plants respond to drought stress. The p44MMK4 kinase of alfalfa is rapidly activated under drought conditions, which is not observed under all abiotic stress conditions. For example, heat shock stress and salt stress fail to activate it. In addition, the MMK4 kinase pathway is independent of abscisic acid (ABA), and its activation mechanism may involves post-translational modifications [16,17].

MYB, NAC, and WRKY are the main transcription factors of alfalfa in response to drought stress, and they help alfalfa adapt to the drought environment by regulating the expression of genes related to stress tolerance. For example, *MsMYBH* can directly promote the expression of two tonoplast-intrinsic aquaporin genes (*MsMCP1*, *MsMCP2*), a peroxidase gene (*MsPRX1A*), a chlorophyll *a-b* binding protein gene (*MsCARCAB1*). By regulating water homeostasis, scavenging excess hydrogen peroxide (H_2_O_2_), and enhancing photosynthesis, it thereby improves the alfalfa’s tolerance to drought [18] (Figure 1). In the response of alfalfa to drought stress, dehydrin MsDHN1 and aquaporin MsPIP2;1 play key roles. Under normal conditions, these two proteins interact with a membrane-anchored MYB protein (mMYB). Under water-deficit conditions, the phosphorylation of MsPIP2;1 causes the C-terminus mMYBΔ83 of mMYB to be released from the plasma membrane into the nucleus. In the nucleus, the C-terminal of MsDHN1 interacts with mMYBΔ83, promoting its transcriptional activity. Furthermore, overexpression of *mMYB* and *mMYBΔ83* regulates the expression of *MsCESA3* and *MsCESA7* by binding to the promoter, thereby enhancing the drought tolerance of transgenic hairy roots. These results suggest that the MsDHN1-MsPIP2;1-MsMYB module plays a crucial regulatory role in drought stress tolerance in alfalfa [19] (Figure 1). The NAC family (NAM, ATAF, and CUC) responds to drought stress by affecting the antioxidant system and osmotic pressure. Notably, the expression of *MsNACs* is significantly upregulated under drought stress [20]. Heterologous overexpression of *MsNAC51* markedly enhances drought tolerance in tobacco (*Nicotiana tabacum* L.) by increasing the accumulation of proline and peroxidase, alongside altering the expression of several drought-responsive genes. Meanwhile, MsNAC51 can directly bind to the promoters of proline synthesizing gene *MsP5CS*, and reactive oxygen scavenging gene *MsPOD-P7*, thereby promoting their expression [21] (Figure 1). In addition, *MsWRKY11* is activated by MsWRKY22 under osmotic stress, serving as a positive regulator of drought tolerance in alfalfa. Overexpression of *MsWRKY11* in plants increases the lignin content of stems under both normal and drought conditions, which may improve water retention and utilization efficiency in response to drought stress [22].

Late embryogenesis abundant (LEA) proteins are important stress response proteins involved in the protection of plants against abiotic stresses. Under drought stress, *MsLEA1* and *MtDehyd* genes were up-regulated and expressed [23,24,25]. Overexpression of *MsLEA1* significantly affects the photosynthetic rate, Rubisco activity, and superoxide dismutase (SOD) activity of alfalfa. Further analysis reveals that MsLEA1 interacted with target proteins such as Fe-SOD and Cu/Zn-SOD, and Rubisco large subunit protein Ms1770, which enhances the adaptability of alfalfa to adverse conditions [24] (Figure 1). MicroRNAs (miRNAs) play important roles in plant growth, development, and stress response. Overexpression of *miR156* in alfalfa enhances root growth and water retention under drought stress, and miR156 also targets *WD40-2* to improve drought tolerance in alfalfa [26] (Figure 1). miR397-5p enhances drought tolerance in tobacco by regulating osmotic and antioxidant enzyme pathways, and the miR397-5p-LAC4 module also plays a role in regulating lignin accumulation in response to drought stress [27] (Figure 1). Furthermore, three genes—*MsNTF2L*, a nuclear transporter gene [23], *MsTHI1*, a thiamine thiazole synthase gene [28], and *MsDIUP1*, a drought-induced unknown protein 1 gene [29]—contribute to enhancing drought tolerance in plants by regulating antioxidant defense system, maintaining osmotic homeostasis and mediating plant signal transduction. Notably, the expression of *MsNTF2L* is highly induced under ABA and drought stress. Its overexpression greatly affects the deposition of cuticular wax, and transcriptome data showed that in both the overexpression (OE) and RNA interference (RNAi) lines of alfalfa, it impacts on a large number of genes involved in wax biosynthesis and transport [23]. MsC3H29, a CCCH-type zinc finger protein, enhances the drought resistance of alfalfa by increasing the accumulation of flavonoids in hairy roots [30].

## 3. Salinity Stress

Salinity stress, a major abiotic stress, impacts alfalfa yield and quality, typically resulting from high salinity and frequently accompanied by high soil pH. Under salinity stress, alfalfa experiences a reduction in seed germination [31], inhibited root growth, particularly during the seedling stage [32,33,34], and alterations in nutritional quality [35,36]. Salinity stress often harms alfalfa through various mechanisms, such as osmotic stress [37], ionic stress [36], oxidative damage [35], and physiological dysfunction [32,38]. Alkali stress is usually accompanied by high pH damage [34], leading to significant inhibition of photosynthesis and nitrogen metabolism [39]. Throughout its evolutionary history, alfalfa has developed strategies to counteract these effects by adjusting the levels of its osmoprotective compounds such as amino acids, saccharides such as pinoresinol, and accumulating lignin content to maintain osmoregulation [37,40]. Alfalfa with strong saline-alkali tolerance also limits the accumulation of sodium ions, maintains the concentration of potassium ions, and enhances the potassium/sodium ratio to sustain ionic homeostasis [41]. Furthermore, it boosts the activities of antioxidant enzymes such as glutathione S-transferase and the accumulates flavonoids and fatty acids to mitigate oxidative damage [40,42,43,44,45]. When salinity and alkalinity co-occur, alfalfa faces even more severe stress, with alkaline stress often outstripping the impact of salinity stress [46,47].

The signal sensing and transduction pathways of alfalfa in response to salinity stress involve a variety of mechanisms such as osmotic and ionic stress-related signaling pathways [48], calcium metabolism [1], long stranded non-coding RNAs [49], phytohormone signaling [50], and flavonoid synthesis [51]. SIMK, a member of the alfalfa MAPK (mitogen-activated protein kinase) family, is involved in mediating the hyperosmotic response to salinity stress and is rapidly and transiently activated [52]. The calcineurin B-like (CBL) family is crucial in plants’ adaptation to abiotic stresses. Specifically, MsCBL4 is instrumental in enhancing tobacco’s salt tolerance by modulating calcium accumulation in roots and maintaining Na^+^/K^+^ ion homeostasis [1]. Long non-coding RNAs (lncRNAs) have been reported to play a significant role in a variety of biological pathways under environmental stress. Specifically, in alfalfa, lncRNAs are involved in phytohormone signaling pathways in response to salt and alkali stress in alfalfa [49]. ABA, a phytohormone, was found to mediate the salt and alkali stress response by inducing the expression of *MsMYB12* and *MsFLS13*, increasing flavonol levels, and maintaining antioxidant homeostasis in alfalfa [33] (Figure 2). In addition, *MsPBL*, a novel stress-responsive gene, may act as a signal transduction protein to allow alfalfa to rapidly respond to the environmental stress signals [53].

Alfalfa enhances its tolerance to salt stress by regulating mechanisms such as the antioxidant system, cell wall structure, calcium metabolism, ion homeostasis, photosynthesis, and nitrogen metabolism. Current research has reported that certain transcription factors and genes play important roles in these processes [43,44,51,54]. WRKY transcription factors are closely related to salinity stress. For example, overexpression of *MsWRKY33* can significantly alter the expression of reactive oxygen species (ROS) scavenger genes. Further studies have found that *MsWRKY33* can directly bind to the promoter region of *MsERF5*, activating its transcription and finely regulating the activity of ROS scavenging enzymes. MsWRKY33 also interacts with the functional domain of MsCaMBP25, a key protein in Ca^2+^ signaling, and this interaction is critical for regulating the plant’s response to salt stress [54] (Figure 2). MsMYB4 enhances salt tolerance in alfalfa under salt stress through an ABA-dependent pathway [55]. The R2R3 MYB transcription factor MsEOBI enhances salt tolerance by promoting flavonoid and lignin biosynthesis and facilitating ROS scavenging. Addtionally, MsEOBI activates the expression of *MsPAL1*, a key gene in the phenylpropanoid pathway, thereby enhancing salt tolerance in alfalfa [56] (Figure 2). Furthermore, the transcription factor MsMYB12 can activate the expression of *MsFLS13*, which affects the antioxidant system and photosynthesis by regulating flavonol synthesis, further enhancing plant tolerance to salt stress [33] (Figure 2). Research has found that the overexpression of transcription factor *MsbZIP53* increases the expression of inositol oxygenase gene *MsMIOX2.* MsMIOX2 activation promotes the biosynthesis of cell wall pectin and hemicellulose, enhances the antioxidant system and photosynthesis, and thereby improves the salinity stress tolerance of alfalfa [44] (Figure 2).

MsRCI2 family is induced to be expressed under salinity stress and plays an important role in the salt tolerance of alfalfa. MsRCI2D and MsRCI2E increase the activity of antioxidant enzymes, reduce the content of ROS, and maintain the balance of Na^+^ and K^+^ ions by regulating the expression of *H^+^-ATPase*, *SOS1*, *NHX1*, *HKT*, which leads to higher salt tolerance in alfalfa [41] (Figure 2). In addition, overexpression of *MsRCI2A*, *MsRCI2B*, and *MsRCI2C* can also improve salt tolerance in alfalfa [57], and overexpression of glutathione s-transferase gene *MsGSTU8* can enhance the resistance of plants to salt and alkaline stress by reducing ROS and increasing the levels of antioxidant enzyme [58].

## 4. Low Temperature Stress

Alfalfa improves cold tolerance by increasing soluble sugars, proteins, proline, and antioxidant capacity (e.g., SOD, catalase (CAT), and flavonoid biosynthesis) under low-temperature stress [4,59], while root type [60], root architecture [61], rhizobial symbiosis [62], and H_2_S signaling [63] also have significant effects on cold tolerance.

Alfalfa’s signal sensing and transduction pathways in response to low-temperature stress mainly involve calcium influx, the MAPK pathway, and the ABA pathway. Under low-temperature conditions, the influx of calcium ions into alfalfa cells was increases, and this process plays a crucial role in cold acclimation. Further studies reveal that this process may be closely related to specific calcium-dependent protein kinases (CDPKs), such as MSCK1 and MSCK2 [64]. Many calcium signaling and CBF/DREB1 pathway-related genes have been found to be strongly affected by cold tolerance according to the transcriptome data [65]. Taken together, these results suggest that calcium signaling plays a critical role in the development of cold tolerance in alfalfa. In addition, under cold stress, the MAP kinase MMK4 in alfalfa is transiently activated by post-translational modification by upstream factors [16]. Low temperature treatment also induces the expression of the *MsPYL* gene, which encodes an ABA receptor protein, indicating that ABA signaling may also play a role in cold stress response [66].

The regulatory network of transcription factors in alfalfa under cold stress mainly involves the bhlh [67,68], NF-Y [69], HSF [6], NAC [70], WRKY [71], and ERF [72] families, which enhance the cold tolerance of plants by regulating gene expression and metabolic pathways such as galactose metabolism, fatty acid metabolism, and glutathione metabolism [73]. ABC transporter proteins are capable of transporting a wide range of substrates and are essential for maintaining cellular homeostasis, and their members in alfalfa. *ABCC8* and *ABCC3*, as well as cold-acclimation-specific (CAS) gene, are critical for the generation of freezing tolerance [65,74]. The calmodulin-like proteins *MsCML10* is induced by cold treatment, and overexpression of *MsCML10* increases the cold tolerance of alfalfa. Further research indicates that MsCML10 can interact with glutathione S-transferase MsGSTU8 and fructose 1,6-biphosphate aldolase MsFBA6. These interactions are dependent on Ca²⁺ and maintain the homeostasis of reactive oxygen species, promoting sugar accumulation for osmotic regulation, respectively [75] (Figure 3). In addition, transcriptome analysis identified two fibronectin (FIB) genes that are key genes in the response of alfalfa to cold stress [76]. The S-adenosylmethionine synthetase (SAMS) gene *MfSAMS1* was significantly induced under cold stress, enhancing the cold tolerance of alfalfa by upregulating polyamine oxidation and antioxidant enzyme activities, thus improving the scavenging ability of H_2_O_2_ under cold stress [77] (Figure 3). The ethylene responsive factor (ERF) gene *MfERF1* improves the cold tolerance of transgenic tobacco by promoting polyamine metabolism, antioxidant protection and proline accumulation. Moreover, it significantly activates the expression of cold-responsive genes (such as *SAMDC1* and *SPMS*), which in turn improves the adaptability of plants to cold stress [78] (Figure 3).

Low temperature stress also causes dynamic changes in metabolic pathways in alfalfa. For example, pathways involved in the synthesis of antifreeze proteins may be activated to increase the cold tolerance of the plant [79]. Cold-tolerant alfalfa slows autotrophic metabolism and biosynthesis and increases protein folding and biosynthesis to withstand cold stress [80]. In addition, cold-tolerant alfalfa seeds exhibit enhanced germination at low temperatures, with genes related to galactose metabolism, fatty acid metabolism, and glutathione metabolism playing important roles in regulating seed germination [73].

## 5. Heavy Metal Stress

Accumulation of heavy metals such as lead, cadmium, copper, chromium, and arsenic accumulate in soil and water have serious effects on plant growth and development. Excesses of these elements result in reduced shoot length, inhibition of plant growth, leaf yellowing, decreased photosynthetic rate, increased oxidative stress, and physiological dysfunction [5,81,82]. Interestingly, lead (Pb) stress has been found to promote root development in alfalfa, with increases in root fresh weight, primary root length, primary root width, and number of lateral roots [83].

Alfalfa responds to stress by inhibiting biosynthesis, signal transduction downstream of growth hormone and ethylene-mediated signaling. Under aluminum stress, it employs internal detoxification mechanisms [84]. Excess Cu^2+^ and Cd^2+^ ions activate distinct MAPK pathways in alfalfa seedlings in response to heavy metal stress. Specifically, Cu^2+^ and Cd^2+^ treatments activate MAPK pathways such as SIMK, MMK2, MMK3, and SAMK in alfalfa roots, with H_2_O_2_ potentially playing a role in the MAPK activation pathway induced by CuCl_2_. In addition, in protoplasts, SIMKK mediated CuCl₂-induced activation of SIMK and SAMK, but does not affect MMK2 and MMK3 or the CdCl₂-induced MAPK activation. These findings suggest that alfalfa can discriminate between different metal stresses, utilizing specific kinase cascade responses to finely regulate its defense mechanisms [85].

Transcription factors such as bHLH115 and MYB can enhance plant tolerance to heavy metals such as cadmium and aluminum, by regulating the uptake of iron and other metal ions, antioxidant responses, and secondary metabolic pathways. Heterologous overexpression of the alfalfa transcription factor *MsbHLH115*, which responds to iron deficiency and cadmium stress, leads to the up-regulation of iron homeostasis-regulated genes, reactive oxygen species-related genes, and metal chelation and detoxification genes in *Arabidopsis thaliana.* This results in increased accumulation of iron and zinc, which helps alleviate cadmium toxicity and enhances cadmium tolerance in *Arabidopsis thaliana*. The MsbHLH115 protein recognizes the E-box element and interacts with the *MsbHLH121* promoter [86] (Figure 4). *MsMYB741* is regulated by the transcription factor MsABF2. The overexpression of *MsMYB741* reduces the aluminum content in alfalfa seedlings, while it also activates the expression of *MsPAL1* and *MsCHI* at the transcriptional level to increase the accumulation of flavonoids in the roots and the secretion from the root tips, thus increasing the aluminum stress resistance of alfalfa [87] (Figure 4).

Specific genes in alfalfa have different response mechanisms to different heavy metal stresses. The expression levels of two metal chelators (*MsPCS1*, *MsMT2*) and two metal transporter protein genes (*MsIRT1*, *MsNRAMP1*) were up-regulated under cadmium stress. Further studies show that the metal transporter protein genes are induced to enhance cadmium transport from roots to shoots, suggesting that they play a key role in cadmium uptake and translocation to the aerial parts of the plant [88] (Figure 4). In addition, both iron (Fe) deficiency and cadmium (Cd) stress induced the expression of *MsYSL6* in alfalfa, resulting in increased Fe accumulation and promoted Cd resistance. This provides potential genetic targets for heavy metal accumulation and phytoremediation [89]. Under aluminum stress, MsLEA1 enhances alfalfa tolerance to aluminum stress by protecting chloroplast structure and function through interactions with iron superoxide dismutase (Fe-SOD) and Cu/Zn superoxide dismutase (Cu/Zn-SOD), as well as the large subunit protein of Rubisco (Ms1770) [24] (Figure 4). Furthermore, MsDHN1 significantly mitigates Al toxicity in alfalfa and is directly regulated by ABA-dependent MsABF2. Two water channel proteins, MsPIP2;1 and MsTIP1;1, may be involved in Al-induced oxalate secretion [90]. In addition, the *MsCOPT* gene in alfalfa is likely to play an important role in the uptake and transport of copper in the roots [91].

## 6. Pathways to Enhance Alfalfa Tolerance to Abiotic Stresses

Microbial symbiosis enhances the tolerance of alfalfa to abiotic stresses, with key contributions from Bacillus [92,93], iron-producing carrier bacteria [94], and Rhizobium [95]. It has been shown that rhizobacteria in symbiosis with alfalfa overproduce indole-3-acetic acid (IAA), leading to alterations in endogenous IAA levels, thereby improving their response to drought stress [95]. In addition, the application of various exogenous substances, such as exogenous proteins, biochar, and hydrogen sulfide, has proven effective in bolstering alfalfa’s resistance to a variety of abiotic stresses [63,96,97,98,99,100,101]. For example, the exogenous harpin protein PopW, derived from the phytopathogenic bacterium *Ralstonia solanacearum*, enhances alfalfa’s adaptation to drought stress by altering IAA levels and upregulating the expression of growth hormone and drought-responsive genes [100]. Additionally, hydrogen-rich water (HRW) can reduce cadmium toxicity in alfalfa seedlings by reducing oxidative damage, enhancing sulfur compound metabolism and maintaining nutrient homeostasis [101].

Genetic engineering has achieved remarkable results in improving the resistance of alfalfa to abiotic stresses. For example, fusion of yeast *TPS1*-*TPS2* genes and their subsequent expression in alfalfa led to the development of transgenic plants that accumulate alginate. This enhancement confers higher tolerance to various stresses such as drought, salinity, freezing, and heat [102]. Overexpression of V-H^+^-ATPase subunits B, C, and H resulted in improved tolerance of transgenic alfalfa to salt stress [103]. In addition, overexpression of the miR156 precursor gene in monocotyledonous rice (*Oryza sativa* L.) not only improved salt and drought tolerance in alfalfa, but also enhanced forage quality [104]. The integration of the sweet potato *orange* gene into alfalfa has led to transgenic plants that exhibit enhanced abiotic stress tolerance and elevated nutritional value [105]. Under field conditions, transgenic alfalfa co-expressing the *NXH* and *H^+^-PPase* genes from the drought-resistant plant, ZxNHX and ZxVP1-1, showed higher net photosynthetic rate, stomatal conductance and water use efficiency, and significantly increased tolerance to high salinity and drought [106]. In addition, transgenic alfalfa plants overexpressing the *Arabidopsis thaliana* ATP sulfurylase gene show increased tolerance to heavy metals and an increased capacity for metal uptake [107]. Furthermore, transgenic alfalfa plants co-expressing glutathione S-transferase (GST) and human P4502E1 (CYP2E1) genes exhibit greater resistance to and accumulation of heavy metal-organic composite pollutants, which contributed to mixed phytoremediation of heavy metal-organic pollutants [108,109]. By overexpressing specific genes in alfalfa, tolerance to abiotic stresses such as salt, drought, heavy metals and organic pollutants can be significantly enhanced, and the quality and nutritional value of the pasture can be improved to protect ecological security.

## 7. Conclusions and Perspectives

Currently, molecular research into how alfalfa responds to abiotic stresses has primarily centered on individual stress factors, yielding significant insights into areas such as signal perception and transduction, as well as transcriptional regulation. Despite these advances, there remain uncharted territories, particularly in the realm of post-translational modifications. Moreover, in cold stress and heavy metal stress, there are relatively few in-depth studies on functional gene regulation mechanisms. Additionally, how alfalfa seeds respond to abiotic stresses during the germination stage, as well as the differences and connections between the response mechanisms at the germination and seedling stages, require further investigation. Future research still needs to identify key genes based on transcriptome, proteome, metabolome and other multi-omics, and combine with molecular biology, biochemistry, cell biology and other research techniques for in-depth mechanistic analysis. Meanwhile, employing a combination of single-cell genomics, in situ spatial genomics and real-time imaging technology to explore the molecular dynamics of alfalfa under abiotic stress will provide theoretical support for the genetic enhancement of alfalfa and the development of stress-resistant cultivars.

## Figures and Tables

**Figure 1 plants-14-00487-f001:**
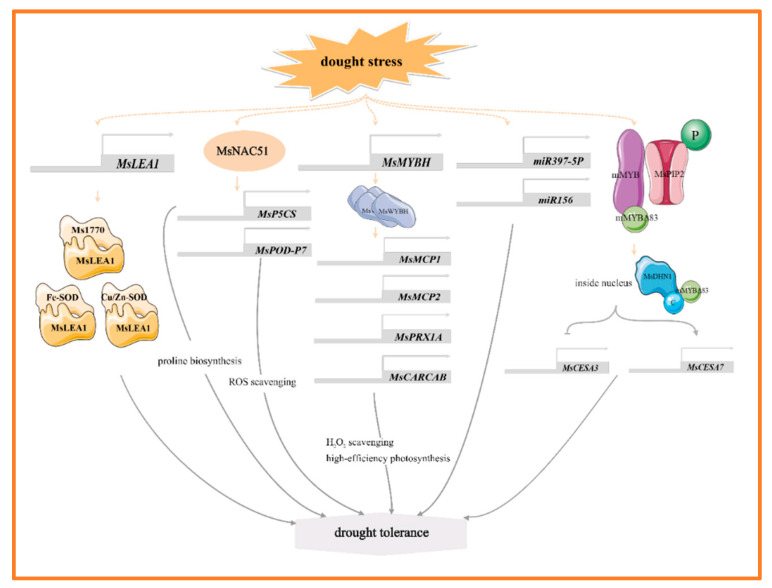
The main molecular mechanism of alfalfa in response to drought stress.

**Figure 2 plants-14-00487-f002:**
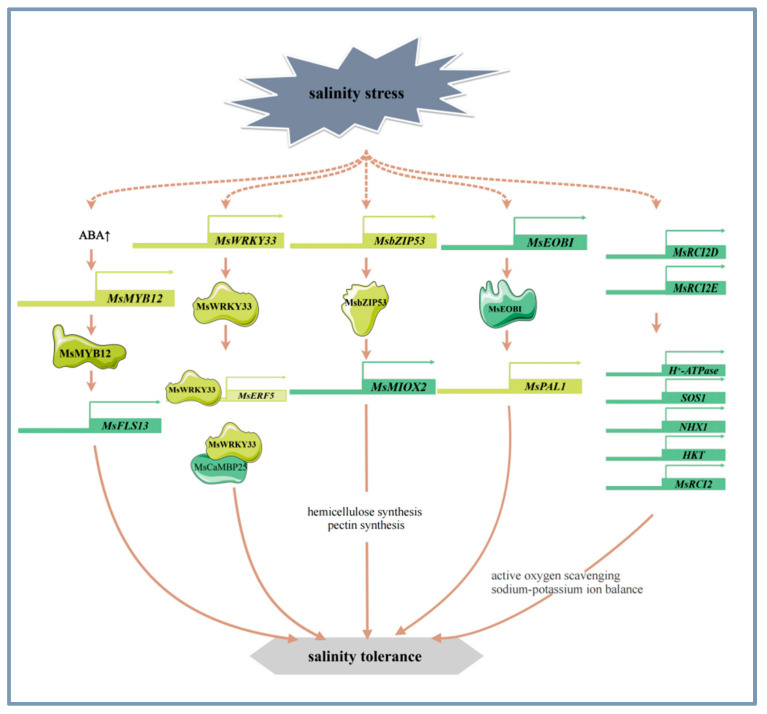
The main molecular mechanism of alfalfa in response to salinity stress.

**Figure 3 plants-14-00487-f003:**
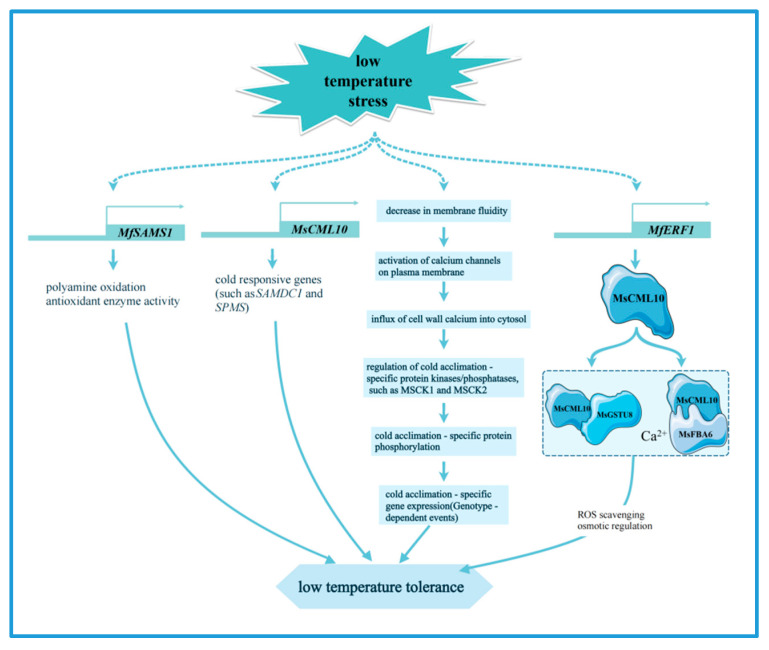
The main molecular mechanism of alfalfa in response to low temperature stress.

**Figure 4 plants-14-00487-f004:**
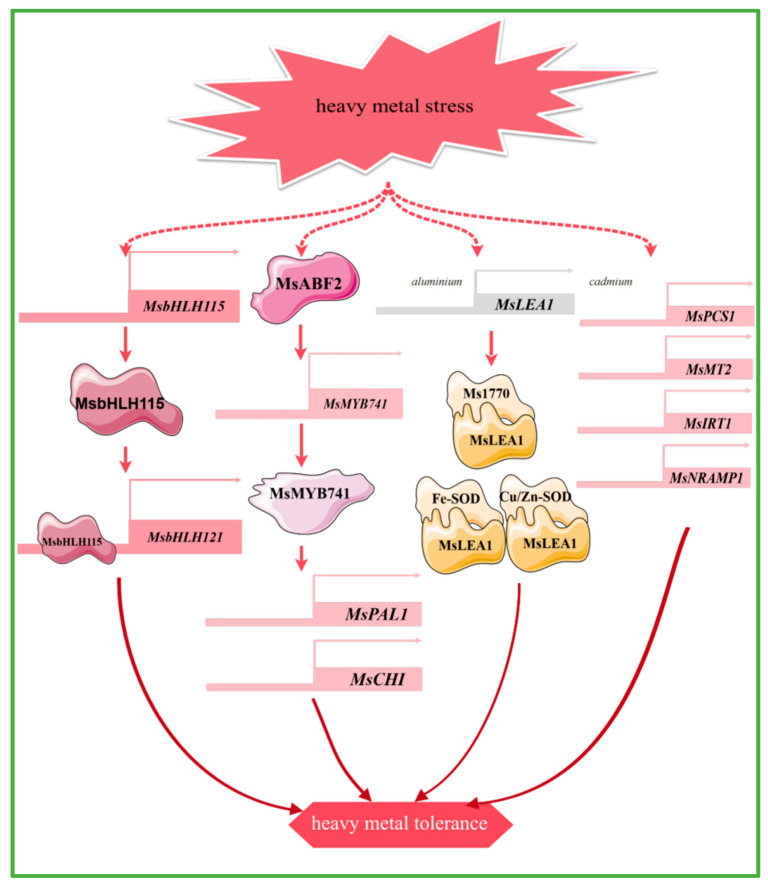
The main molecular mechanism of alfalfa in response to heavy mental stress.

## Data Availability

No new data were created or analyzed in this study. Data sharing is not applicable to this article.

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
