# Peer review of "Molecular Mechanisms of Alfalfa Response to Abiotic Stresses"

_plants, 2025, doi:10.3390/plants14030487_

Round 1
Reviewer 1 Report
Comments and Suggestions for Authors
The article is very well written and organized and gives a great overview of molecular control of alfalfa under abiotic stresses. I have some very minor suggestions. Check that acronyms are indicated first time I found a few not introduced ROS, SOD, CAT, etc. Indicate binomial Latin name of all plants and microorganisms
Minor revisions:
l. 19 North
l. 55 sentence unclear. what does ‘with some specificity’ means
l. 76 tobacco add binomial (Nicotiana tabacum L.)
l. 88 indicate what SOD means? first time indicated in text
l. 177 indicate what is CAT
l. 187 calcium, small case
l. 297 rice, (Oryza sativa L.)
Revise reference list and Italicize binomial names in references. Write Genus species, some are all in small case letters l. 360, 564, 590, 597
Although you have many references and the article is well supported by published research, most references are from authors from China. I am not criticizing Chinese researcher; you do excellent work; I am just wandering if there were no more articles from research in Europe or North America.
Great article looking forward to see it published!
Reviewer 2 Report
Comments and Suggestions for Authors
The manuscript (plants-3427668) submitted by Peng et al. summarizes the recent advances in the molecular mechanism of alfalfa plants in response to four major abiotic stresses. It is a timely addition to the existing review library. I believe readers will be surely interested in citing this for their future research in alfalfa improvement. I just have a few minor comments for the authors to address:
1. Line 68. "the C-terminal" should be "the C-terminus";
2. Line 113. "harm" should be "harms";
3. Line 183. "reveales" should be "reveal";
4. Line 185. "Calcium" should be "calcium";
5. Line 395-396. Reference 26 is not in the right format. All other references use abbreviated journal names, but not this one.;
6. Some references are too old.
Comments on the Quality of English LanguageOverall the English in the entire manuscript is good. Some minor typos/misuses are indicated in the comments to authors.
Reviewer 3 Report
Comments and Suggestions for Authors
extremely well written. However, I think a couple major areas are unaddressed:
1) why is germination affected differently than growth?
2) no mention is made of accumulation of tertiary compounds, which are a key component of abiotic stress tolerance
